# Glucosylceramides from *Cladosporium* and Their Roles in Fungi–Plant Interaction

**Mariana Ingrid Dutra da Silva Xisto** [1,†] , **Mariana Collodetti Bernardino** [1,†] , **Rodrigo Rollin-Pinheiro** [1,†] , **Caroline Barros Montebianco** [2,†] , **Andrêina Paula da Silva** [1] , **Renata Oliveira Rocha Calixto** [1] , **Bianca Braz Mattos** [1] , **Maite Freitas Silva Vaslin** [2] and **Eliana Barreto-Bergter** [1,*]

[1] Laboratório de Química Biológica de Microorganismos, Instituto de Microbiologia, Universidade Federal do Rio de Janeiro (UFRJ), Av. Carlos Chagas Filho, 373, CCS, Rio de Janeiro 21941590, Brazil; marylanax@gmail.com (M.I.D.d.S.X.); mcollodetti@gmail.com (M.C.B.); rodrigorollin@gmail.com (R.R.-P.); andreinapaula255@gmail.com (A.P.d.S.); rorcalixto@gmail.com (R.O.R.C.); biabmattos@hotmail.com (B.B.M.)
[2] Laboratório de Virologia Molecular Vegetal, Departamento de Virologia, Instituto de Microbiologia, Universidade Federal do Rio de Janeiro (UFRJ), Av. Carlos Chagas Filho, 373, CCS, Rio de Janeiro 21941590, Brazil; cbmontebianco@yahoo.com (C.B.M.); maite@micro.ufrj.br (M.F.S.V.)
\* Correspondence: eliana.bergter@micro.ufrj.br
† These authors contributed equally to this work.

**Abstract:** *Cladosporium* species are widely distributed filamentous fungi. One of the most important species is *C. herbarum*, which is related to infections in a variety of plants and of concern in plantations. Fungal cerebrosides, such as glucosylceramide (GlcCer), have been described as playing important roles in fungal growth and pathogenesis, but GlcCer from *C. herbarum* has not been characterized so far. For this reason, the present study aimed to elucidate the chemical structure of GlcCer from *C. herbarum* and its role in the interaction with *Passiflora edulis*. Mass spectrometry characterization of purified GlcCer revealed two major molecular ions, *m*/*z* 760 and *m*/*z* 774, and it reacts with monoclonal anti-GlcCer antibodies and is exposed on the fungal surface. *P. edulis* treatment with GlcCer induced increased levels of superoxide as well as the expression of some genes related to plant defense, such as *PR3*, *POD*, *LOX* and *PAL*. GlcCer also enhanced growth parameters, such as plant height and root weight. All these results suggest that *C. herbarum* GlcCer can stimulate plant defense mechanisms, which could help plants to face fungal infections.

**Keywords:** glucosylceramides; *Cladosporium herbarum*; *Passiflora edulis*; plant defense-related genes

## 1. Introduction

*Cladosporium* species are hyaline filamentous fungi widely distributed in nature [1]. Among this group, two species are most important: *Cladosporium resinae* and *Cladosporium herbarum*. *C. resinae* is known to degrade biofuel and to contaminate storage tanks [2]. *C. herbarum* is an important plant pathogen that infects passion fruit, peanut, potatoes, oats, wheats, onions, grapes and coffee [3].

The fungal cell wall is an important structure that plays a role in fungal communication with the environment, protection against stress and the maintenance of cell morphology [4]. A variety of glycoconjugates on fungal surfaces have already been described. Glycosphingolipids, such as glucosylceramides (GlcCers), are cerebrosides composed of a fatty acid linked to a sphingoid base, and a sugar moiety that is usually glucose or galactose [5]. This structure is considered conserved among fungi, but some variations are observed in the fatty acid moiety, which include the degree of saturation and carbon chain length. Glucosylceramide from *C. resinae* has already been described, consisting of a ceramide of a 9-methyl-4,8-sphingadienine, which is linked to a saturated fatty acid with 16 carbon lengths [6]. However, glucosylceramides from *C. herbarum* have not been characterized so far.

Fungal cerebrosides have already been reported to play a role in fungi–plant interaction, especially in the context of plant diseases caused by fungal pathogens. It has already been demonstrated that cerebrosides from *Fusarium* species are elicitors of the plant defense response against *Fusarium* infection [7,8]. More recently, glucosylceramides from *Fusarium oxysporum* were shown to induce plant resistance against infection by *Tobacco mosaic virus* (TMV) [9], indicating that fungal sphingolipids stimulate plant defense mechanisms against other pathogens, such as viruses.

The present study aimed at characterizing glucosylceramide from *C. herbarum* and to compare its structure with that from *C. resinae*, as well as to evaluate its exposition on the fungal cell surface. In addition, the role of glucosylceramides on the induction of *Passiflora edulis* defense mechanisms is also analyzed since *C. herbarum* is an important pathogen that infects *P. edulis*.

## 2. Materials and Methods

### 2.1. Microorganism and Culture Conditions

*Cladosporium herbarum* was kindly supplied by Dr. J. Guarro, Unitat de Microbiologia, Facultat de Medicina e Institut d'Estudis Avançats, Réus, Spain. It was grown in Erlenmeyer flasks containing 200 mL of potato dextrose broth (PDB, Acumedia, Lansing, MI, USA). Cultures were incubated at room temperature for seven days with shaking (pre-inoculum). Conidia were grown on Petri dishes containing Potato Dextrose Agar (PDA) medium at room temperature. After seven days, conidia were obtained by washing the agar surface with PBS and hyphal fragments and debris were removed by filtration through 40 µm cell strainer (Falcon®, Corning, NY, USA). Mycelium was obtained by inoculating fungus in Erlenmeyer flasks containing potato dextrose broth, followed by incubation for 7 days at room temperature with shaking.

*Cladosporium resinae* (American Type Culture Collection—ATCC 22712) and *Fusarium oxysporum* (IOC 4247—supplied by Maria Inês Sarquis from the Collection Culture of Institute Oswaldo Cruz, Rio de Janeiro, RJ, Brazil) were grown under the same conditions as *C. herbarum*.

### 2.2. Extraction and Purification of Glycosphingolipids

Total lipids from *C. herbarum*, *C. resinae*, and *F. oxysporum* mycelia were extracted at room temperature using chloroform/methanol 2:1 ($v/v$) and 1:2 ($v/v$), successively, as described by [9]. Extracted lipids were partitioned with chloroform/methanol/0.75% KCl (8:4:3 $v/v/v$), as described by Folch et al. (1957) [10]. Neutral lipids found in the lower Folch layer were fractionated on a silica gel column, which was eluted with chloroform, acetone and methanol. Fractions containing glycosphingolipid were loaded on a new silica gel column and eluted with chloroform/methanol with increasing polarity (9:1, 8:2, 7:3 and 1:1 $v/v$) [11]. The glycosphingolipids recovered after purification were monitored by thin layer chromatography (TLC) on silica gel 60 plates developed with chloroform/methanol/2 M ammonium hydroxide, 40:10:1 ($v/v/v$). The spots were visualized with iodine and orcinol/$H_2SO_4$ [12,13].

### 2.3. Sugar Analysis

To assess the monosaccharide components, GlcCer species were hydrolyzed with 3 M trifluoroacetic acid at 100 °C for 3 h, as described by Calixto et al. (2016) [14]. Monosaccharides were detected by TLC in n-butanol/acetone/water (4:5:1 $v/v/v$) and visualized using the orcinol–sulfuric acid reagent [14]. Reference sugars (rhamnose, glucose, mannose and galactose) were used for sugar identification of *C. herbarum* GlcCer.

### 2.4. ESI-MS Analysis

GlcCer species were analyzed by electrospray ionization (ESI-MS) in positive (ESI+) ion mode, using an ESI-ion Trap instrument (Model Amazon SL, Bruker, Germany). GlcCer species were dissolved in chloroform/methanol/water (5:4:1 $v/v/v$), containing 1 mM

lithium chloride and analyzed via direct injection using a microsyringe pump (Hamilton) [14]. Nitrogen was used as nebulizer and carrier gas.

### 2.5. Reactivity of GlcCer with Monoclonal Anti-Glccer Antibodies (Mabs)

The reactivity of *C. herbarum, C. resinae* and *F. oxysporum* GlcCer to anti-GlcCer Mabs was evaluated by ELISA as described by Nimrichter et al. (2005) [15]. Briefly, GlcCer from these fungal species were dissolved in ethanol–methanol 1:1 (*v/v*), and 1 μg/well was added to a flat-bottomed polystyrene microtiter plate (BD-Falcon, San Jose, CA, USA). After adding GlcCer to the wells, the plate was dried and blocked with PBS containing 1% BSA (2 h, 37 °C). Antibodies anti-GlcCer were serially diluted and an unrelated IgG was added as a negative control. Then, the plate was incubated at 37 °C for 1 h and washed three times. HRP-conjugated anti-mouse IgG (1:1000 dilution) (Sigma-Aldrich, St. Louis, MI, USA) was used as a secondary antibody and incubated for 1 h at 37 °C. The plate was washed three times with PBS and the antigen–antibody complexes were detected with 0.04% ortho-phenylenediamine (OPD) in phosphate–citrate buffer at pH 5.0 containing 30 vol. $H_2O_2$. Absorption at 490 nm was measured using a spectrophotometer (Bio-Rad, Hercules, CA, USA).

### 2.6. Immunolocalization of GlcCer on the Surface of C. herbarum and C. resinae

Mycelial or conidial forms of *C. herbarum* and *C. resinae* were fixed in 4% paraformaldehyde cacodylate buffer (0.1 M, pH 7.2) for 1 h at room temperature. Fixed cells were washed twice in PBS and then incubated in PBS containing 1% BSA for 1 h at 37 °C. Conidia and mycelium were washed three times with PBS and incubated for 1 h at 37 °C with either anti-GlcCer Mab or an isotype-matched control, which was used at a concentration of 50 μg/mL in PBS containing 1% BSA. Cells were washed and incubated in 100 μL of Alexa fluor 546 (at 1:400 dilution) in PBS containing 1% BSA for 1 h at 37 °C. After three washes, the cells were suspended in 50 μL of a mounting solution containing 0.01 M N-propyl gallate diluted in PBS–glycerol (1:1, *v/v*). Ten microliters of the suspension were applied to a microscope slide and examined with an Olympus AX70 fluorescence microscope (Olympus America Inc., Center Valley, PA, USA) using a 620-nm filter and a 100X magnification lens [16].

### 2.7. Plant Culture

Seeds of *Passiflora edulis* were germinated in plant substrate (Garden Plus, Turfa Fertil Co., Criciúma, SC, Brazil) supplemented with vermiculite (1:1) and kept in a greenhouse at the Universidade Federal do Rio de Janeiro (UFRJ), under tropical area natural light conditions with controlled air temperatures of 27 +/− 2 °C. Treatments were performed in plants with three to four true leaves [9,17].

### 2.8. GlcCer Pulverization on P. edulis

The experiment was performed in young passion fruit plants (*P. edulis*) that were sprayed with GlcCer (100 μg/mL) in potassium phosphate buffer (0.01 M, pH 7.0) or only potassium phosphate buffer as a control (five plants for each condition). Suspensions were sterilized by filtration through a 0.20 μm cellulose acetate filter (Advantec®, Dublin, CA, USA) before use. Plants without any treatment were also used as a control. The experimental design was randomized with five plants for each treatment, and two independent experiments were performed [9]. The passion fruit plants were sprayed once and the leaves were collected after 6 h, 24 h, 96 h and 15 days for gene expression evaluation or for histochemical detection of reactive species oxygen (ROS). The volume per spray for each young plant was about 5 mL. The plant developmental evaluation was performed as described for the gene expression evaluation and ROS detection.

### 2.9. Histochemical Detection of Reactive Species Oxygen (ROS)

Superoxide radical production in leaves of *P. edulis*, after GlcCer or buffer pulverization, was evaluated by histochemical test, and superoxide radicals were detected using a solution

containing the nitro blue tetrazolium reagent (NBT, Sigma-Aldrich) at a concentration of 0.5 mg/mL. The leaves were collected 6 h, 24 h and 96 h after pulverization. Untreated plants were used as a control. The leaves were dipped in Petri dishes containing NBT solution and kept in a vacuum chamber (400 bar) in the dark for one hour. The leaves were then boiled in absolute ethanol for chlorophyll removal to provide a better visualization of the precipitates formed by NBT [9,18].

### 2.10. Evaluation of Gene Expression by Real-Time RT-qPCR

Leave samples of *P. edulis* were collected at 6 h, 24 h, 96 h and 15 days after GlcCer or buffer pulverization (five plants each), grouped in a pool of 2 leaves for each independent plant (five plants). Samples were immediately frozen in liquid nitrogen and stored at −80 °C until use. For RNA extraction, leaves were crushed in liquid nitrogen until the formation of a powder that was transferred to an Eppendorf tube and maintained at −80 °C for subsequent extraction of total RNA [9,17].

Expression of four defense-related genes was evaluated in leaves after GlcCer or buffer pulverization: *PR3* (chitinase), *PAL* (phenylalanine ammonia lyase), *LOX* (lipoxygenase) and *POD* (Peroxidase). Total RNA was extracted with TRIzol® reagent (Thermo Fisher Scientific, Waltham, MA, USA). SuperScript™ VILO™ MasterMix (Invitrogen, Waltham, MA, USA) and PowerUp™ SYBR™ Green Master Mix (Thermo Fisher Scientific, Waltham, MA, USA) were used for cDNA synthesis and qPCR reactions, respectively. Two house-keeping genes, ERS (Ethylene Response Sensor) and NDID (NADP-dependent isocitrate dehydrogenase), were used for qPCR normalization. Oligonucleotides described previously by Munhoz et al. (2015) were used. qPCR was performed in technical triplicates and the signal was detected by the AriaMx Real-time PCR System (Agilent Technologies, Santa Clara, CA, USA). The thermocycling conditions were as follows: initial denaturation at 95 °C for 2 min, 40 cycles of 95 °C for 15 s, 55–68 °C (Table S1) for 30 s and elongation at 72 °C for 1 min. Ct values were evaluated using the $2^{-\Delta\Delta Ct}$ method described by Livak and Schmitgen (2001), and represented as relative expression. The fluorescence threshold value was calculated using Agilent Aria 1.8 software (Agilent Technologies, Inc., Santa Clara, CA, USA) [19,20].

### 2.11. Plant Developmental Parameter Evaluation

Forty days after GlcCer or buffer pulverization, plant height and root weight were evaluated. Either fresh or dry (after 48 h at 65 °C) weight of roots was determined. For these evaluations, five individual plants after GlcCer or buffer pulverization, or plants without any treatment, were analyzed, following the non-destructive method described by Souto et al. (2017) [21]. Fresh and dry weights were determined using the accurate electronic scale JA3003N, from Bioprecisa Co, Curitiba, PR, Brazil.

### 2.12. Statistical Analysis

Statistical analysis of all experiments was performed using the Student *t*-test. Data were plotted using GraphPad Prism v5.0 (San Diego, CA, USA) to generate the graphs.

## 3. Results

### 3.1. Structural Characterization of Glycosphingolipids from C. herbarum

A scheme of glycosphingolipids' (GSLs') extraction and purification is shown in Figure 1. Chemical characterization of purified GSLs from C. herbarum was performed using electrospray ionization mass spectrometry (ESI-MS). Two major lithiated, singly charged ion species at *m/z* 760 and *m/z* 776 were observed in the MS1 spectrum (Figure 2). The spectrum (MS2) obtained by collisional activation of the lithium-cationized GSL from C. herbarum (*m/z* 760) is shown in Figure 3A. Fragments were observed at *m/z* 742 (loss of water from [M+Li+]+) and 598 (fragment Yo attributed to loss of hexose) [22]. The latter was accompanied by the corresponding Zo fragment at *m/z* 580, respectively. The prominent ion at *m/z* 480 is originated by cleavage of the amide N-CO bond [22]. The *m/z* increment of 26

between 562 and 536 is consistent with the double bond of 2-hydroxyoctadecenoic acid. An identical collision spectrum was obtained from the lithium-cationized GlcCer of Aspergillus fumigatus [23] and Fusarium solani [22], which has been shown by NMR spectroscopy to be N-2′-hydroxyocatadecenoyl-1-O-hexopyranosyl-9-methyl-4,8-sphingadienine. The MS-MS spectrum of ion species at $m/z$ 776 is shown in Figure 3C. The loss of 162 units, common to GSL, suggested the presence of a monosaccharide unit and gave rise to daughter ions at $m/z$ 614 [M-hexose+ Li+]+, which correspond to the ceramide monolithiated ions from the parent ions at $m/z$ 776. The daughter ion at $m/z$ 496 is consistent with a loss of an OHC18 fatty acid. The difference of 16 units observed at $m/z$ 760 [M+Li+]+ and detectable after the loss of either monosaccharide or fatty acid, suggests that the long chain base could possibly present an extra hydroxyl group. The expression of a GSL consisting of a glucosyl 9-methyl-4,8-hydroxy-sphingadienine carrying an extra hydroxyl group positioned at C-4 or C-5 of the long chain base was observed in sclerotic cells of Fonsecae pedrosoi [15] and *Lyophyllum* sp. strain Karsten and *Trichoderma asperellum* 302 [24], and more recently in *F. oxysporum* [9]. The position of an extra hydroxyl group is unspecified in the scheme (Figure 3D).

Hydrolysis of the glycosphingolipid was performed to elucidate its sugar composition. TLC analysis showed the presence of glucose as the monosaccharide group found in the glycosphingolipid structure (Figure 2), suggesting that these structures correspond to glucosylceramides (GlcCers).

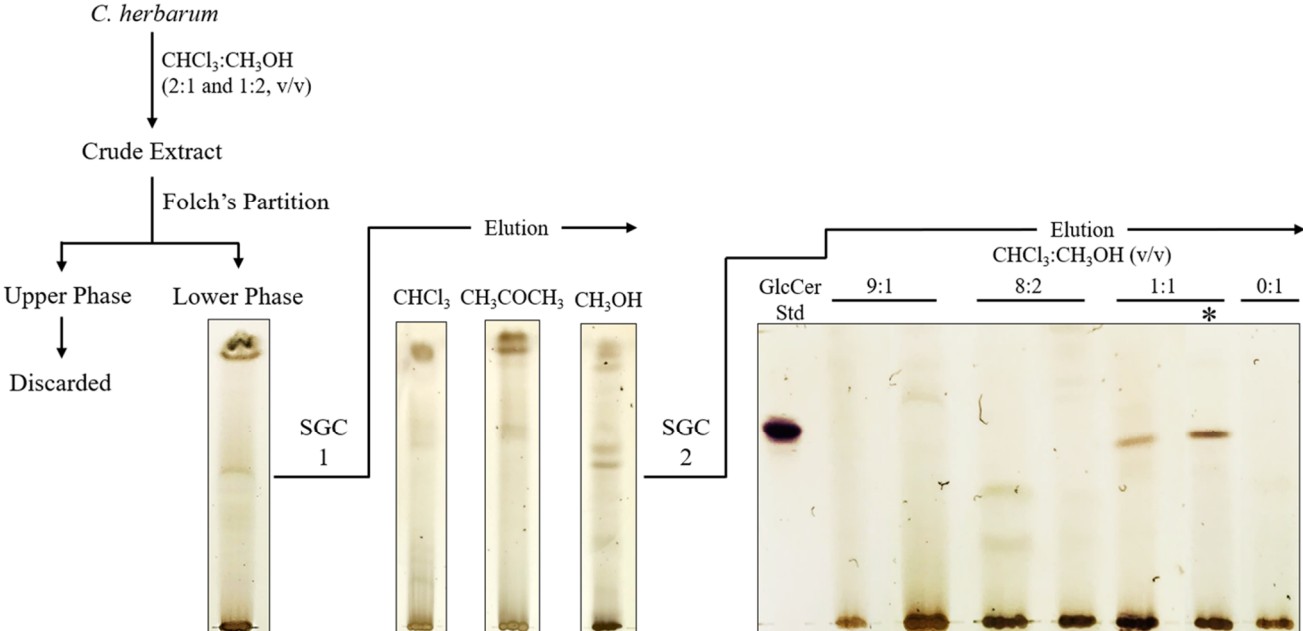

**Figure 1.** Scheme of the isolation and purification of glycosphingolipid from *C. herbarum*. High performance thin layer chromatography (HPTLC) of fungal neutral glycosphingolipids isolated by silica gel column chromatography (SGC). Solvent system for SGC 1: $CHCl_3$, $CH_3COCH_3$ and $CH_3OH$; for SGC 2: $CHCl_3:CH_3OH$ (9:1; 8:2; 1:1 and 0:1 $v/v$). Running solvent for HPTLC–chloroform/methanol 2 M $NH_4OH$ (40:10:1 $v/v/v$). Detection: iodine and orcinol–sulfuric acid reagents. * Purified glycosphingolipid fraction eluted with $CHCl_3:CH_3OH$ 1:1 $v/v$ was analyzed and used in all the experiments.

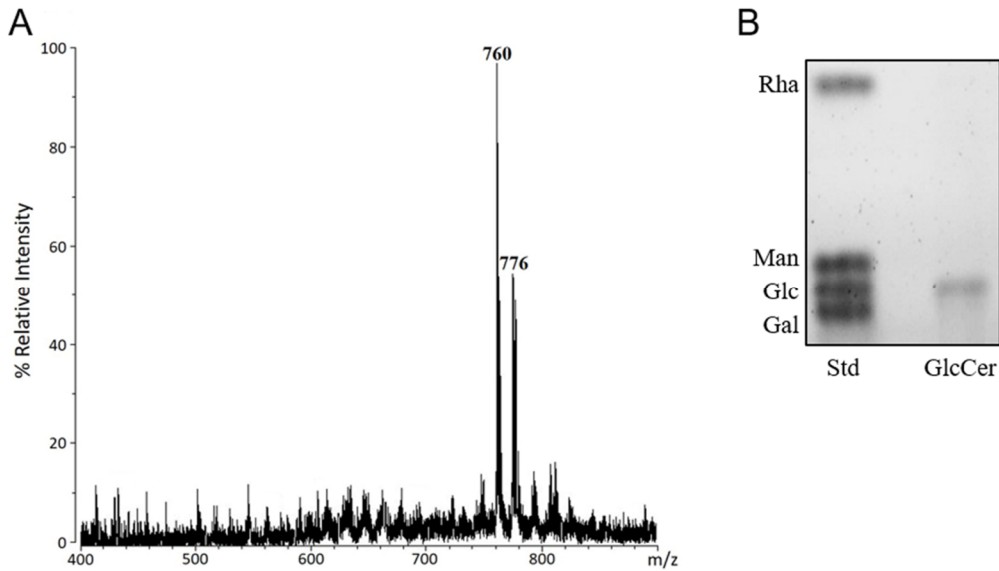

**Figure 2.** Positive ESI-MS (electrospray ionization mass spectrometer) (M+Li) analysis of GlcCer species from *C. herbarum*. ESI-MS1 of GlcCer species from *C. herbarum* (**A**). Thin layer chromatography (TLC) of hydrolyzed sugar from GlcCer species (**B**).

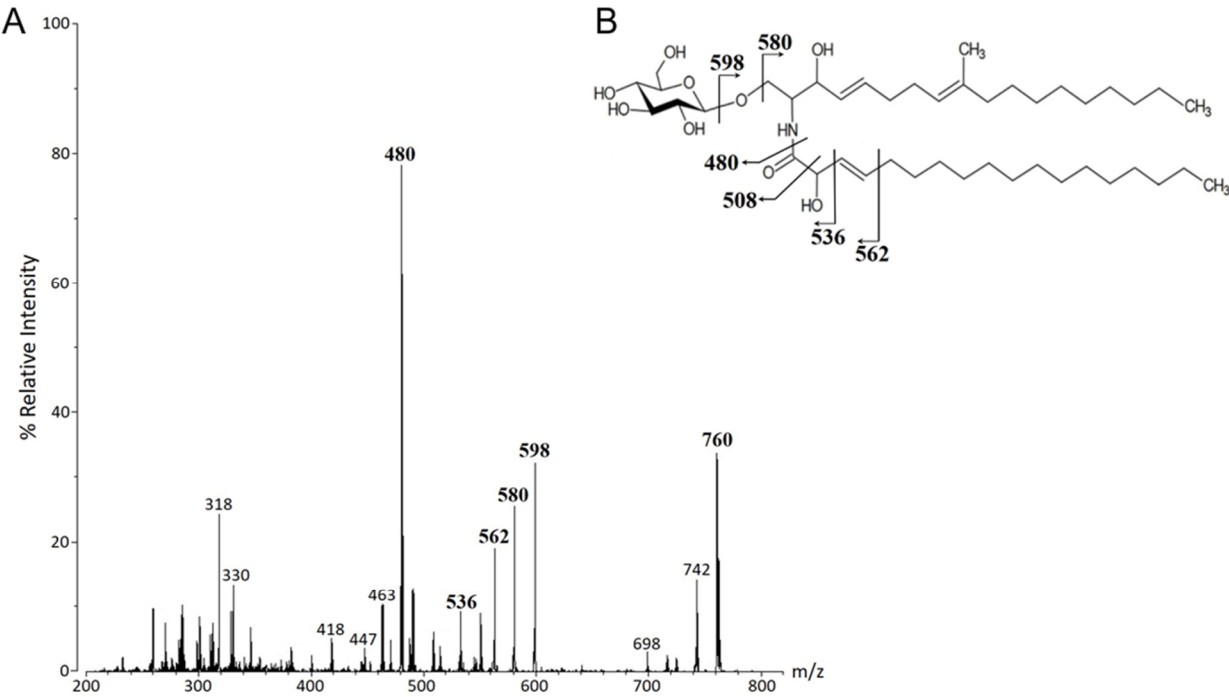

**Figure 3.** *Cont*.

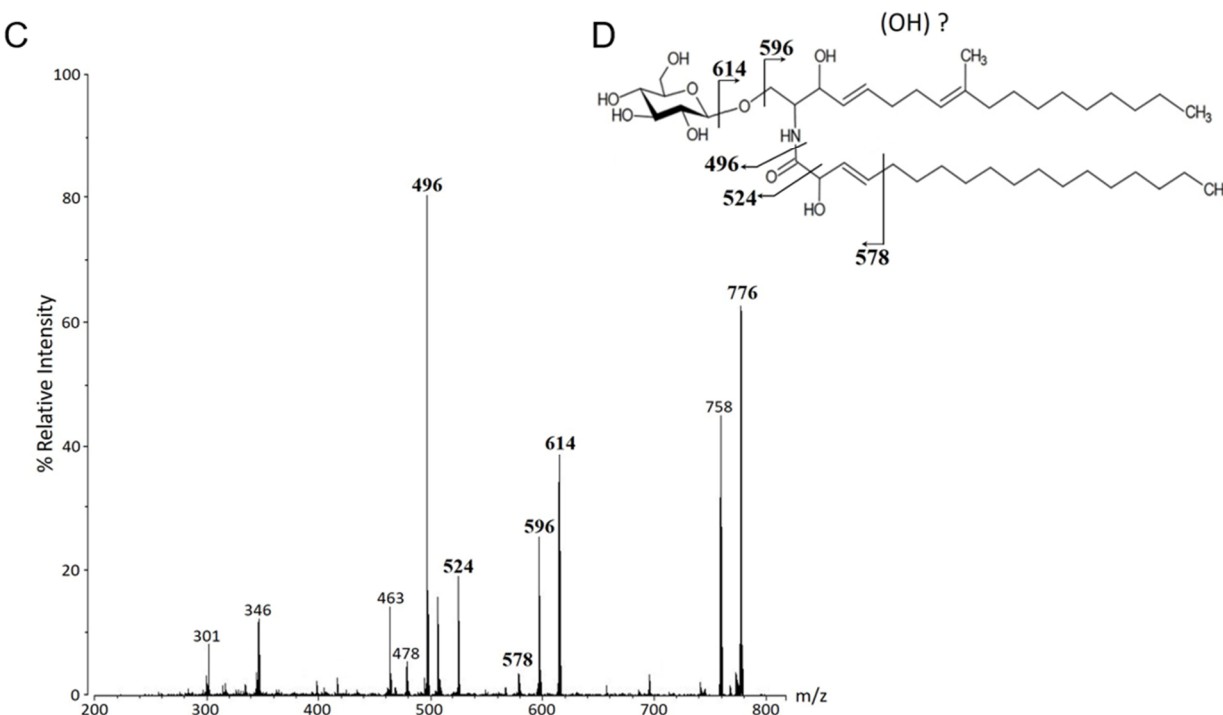

**Figure 3.** ESI-MS2 of the major ion species *m/z* 760 (**A**) and its proposed structure (**B**). ESI-MS2 of the major ion species *m/z* 776 (**C**) and its proposed structure (**D**). The most abundant fragments in each spectrum are in boldface. The position of an extra hydroxyl group of unspecified location is in parenthesis (**D**).

### 3.2. Reactivity of GlcCer with Monoclonal Antibodies (Mabs) Anti-GlcCer

Reactivity of purified *C. herbarum* and *C. resinae* GlcCer to Mab anti-GlcCer was evaluated by ELISA. Both GlcCer reacted with Mabs, similar to that observed for *F. oxysporum* GlcCer (used as a positive control), suggesting that those three GlcCer share the same epitope recognized by the antibody (Figure 4). GlcCer from both *C. herbarum*, *C. resinae* and *F. oxysporum* did not react with irrelevant IgG, indicating that the reactivity was a result of specific linkage between anti-GlcCer Mab and these lipidic structures.

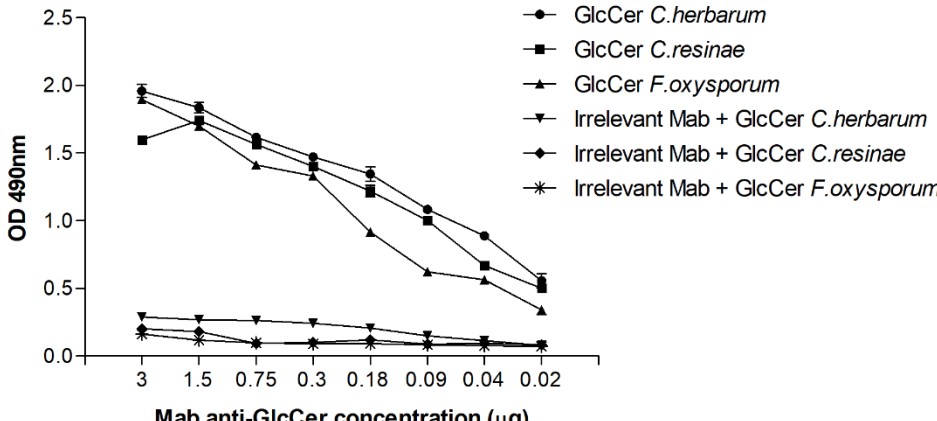

**Figure 4.** Reactivity of anti-GlcCer Mab with GlcCer from *Cladosporium* species, measured by ELISA. *F. oxysporum* GlcCer was used as positive control and irrelevant IgG Mab was used as negative control. The ELISA was performed in triplicate.

### 3.3. Immunolocalization of GlcCer on the Surface of C. herbarum

To check whether GlcCers from *C. herbarum* and *C. resinae* are present on the fungal cell surface, immunofluorescence microscopy using anti-GlcCer Mab was carried out. The results showed that GlcCer is exposed on the surface of the conidia and hyphae of *C. herbarum* and *C. resinae* since both cell types were stained by the monoclonal antibody (Figure 5). The images demonstrate that fluorescence is observed around the cells, suggesting that GlcCer is present on the fungal cell surfaces not only in spores (conidia), but also in the filamentous stages of growth (hyphae).

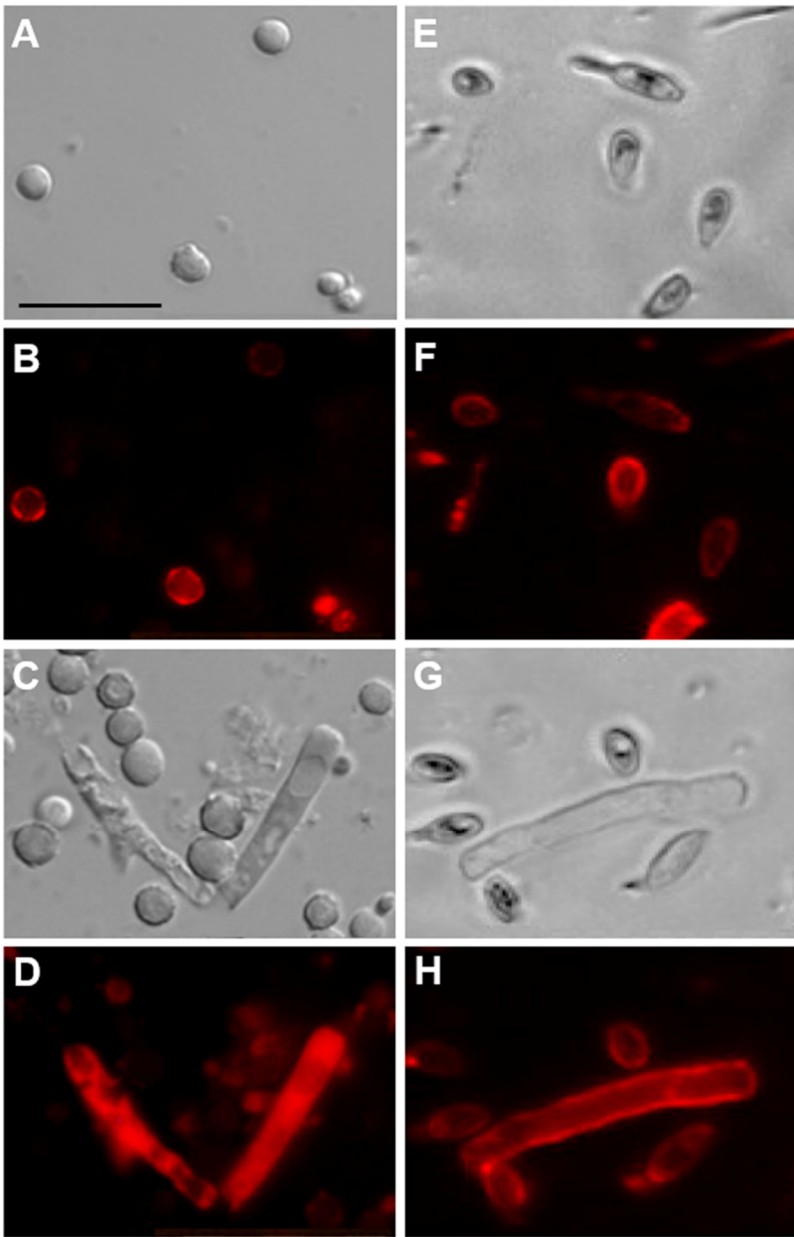

**Figure 5.** Distribution of GlcCer species on *Cladosporium* species' surface. The anti-GlcCer Mab revealed the presence of GlcCer species on the surface of conidia and mycelium from *C. herbarum* (**B**,**D**) and *C. resinae* (**F**,**H**). (**A**,**C**,**E**,**G**) show the images of differential interferencial contrast microscopy (DIC). Bar: 20 µm.

### 3.4. Histochemical Detection of Reactive Oxygen Species (ROS) in P. edulis after GlcCer Treatment

In order to evaluate the role of the *C. herbarum* GlcCer treatment in eliciting the defense mechanisms of *P. edulis*, the presence of superoxide radicals was analyzed since

it is suggestive of plant defense mechanisms. To evaluate whether GlcCer could lead to ROS production, plants of *P. edulis* were sprayed with 100 μg/mL GlcCer and the presence of superoxide radicals was analyzed at 6, 24 and 96 h after treatment. The histochemical analyses were performed using leaf samples previously treated with GlcCer or the control buffer and incubated with NBT solution. The accumulation of superoxide ions ($O_2^-$) is evidenced by violet areas on the leaves. Untreated leaf samples were also used as negative controls. GlcCer-treated leaves showed violet areas, suggesting that superoxide radicals were increased compared to control leaves where violet spots were not observed (Figure 6). These data suggest that GlcCer induces the production of superoxide radicals in all time points analyzed, which represents an important mechanism of defense in plants.

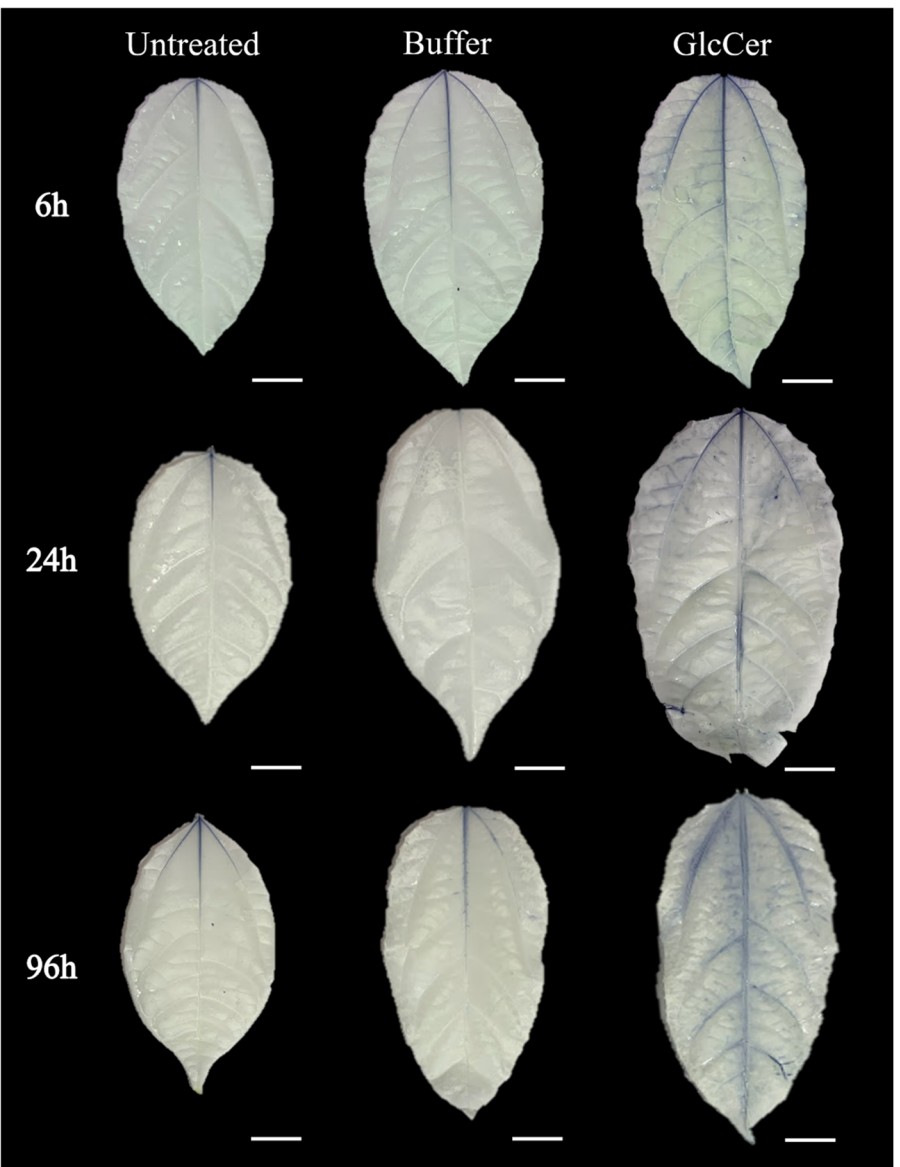

**Figure 6.** Histochemical visualization of reactive oxygen species (ROS) accumulation in leaves of *P. edulis* collected at 6, 24 and 96 h after treatment with *C. herbarum* GlcCer. The accumulation of $O_2^-$ is evidenced by the violet color (NBT precipitation). Bar: 2 cm.

### 3.5. Evaluation of Gene Expression

To evaluate the effect of *C. herbarum* GlcCer treatment in inducing defense mechanisms in *P. edulis*, some genes related to plant defense were analyzed using RT-qPCR (Figure 7). To evaluate the induction of superoxide radicals, the plants of *P. edulis* were sprayed with

100 µg/mL GlcCer, and the defense-related genes were analyzed 6 h, 24 h, 96 h and 15 days after treatment.

The gene *PR3*, which encodes for chitinase, was 4.5- and 28.6-fold more expressed after treatment for 6 h and 15 days, respectively (Figure 7A). Regarding the gene *POD*, which encodes for peroxidase, its expression increased 10.8-fold after 6 h, reducing to only 3.2 fold after 24 h of incubation (Figure 7B). At longer periods, *POD* did not show any difference compared to the control. The expression of the gene *LOX*, which encodes for lipoxygenase, was 9.0- and 13.4-fold increased after 6 and 24 h, respectively (Figure 7C).

Regarding the gene *PAL*, which encodes for phenylalanine ammonia lyase, expression was around 4-, 40- and 112-fold increased after 6 h, 96 h and 15 days, respectively (Figure 7D). After 96 h of incubation, higher levels of *PAL* expression were not observed when compared to the control. These data suggest that GlcCer induces *PAL* production first at earlier times and then again at late periods after GlcCer stimulation.

These results showed that the treatment with *C. herbarum* GlcCer induced higher expression levels of some genes related to plant defense. This response was especially observed after 6 and 24 h, with the exception for *PR3* and *PAL* where a higher level of expression was seen after 15 days of GlcCer treatment.

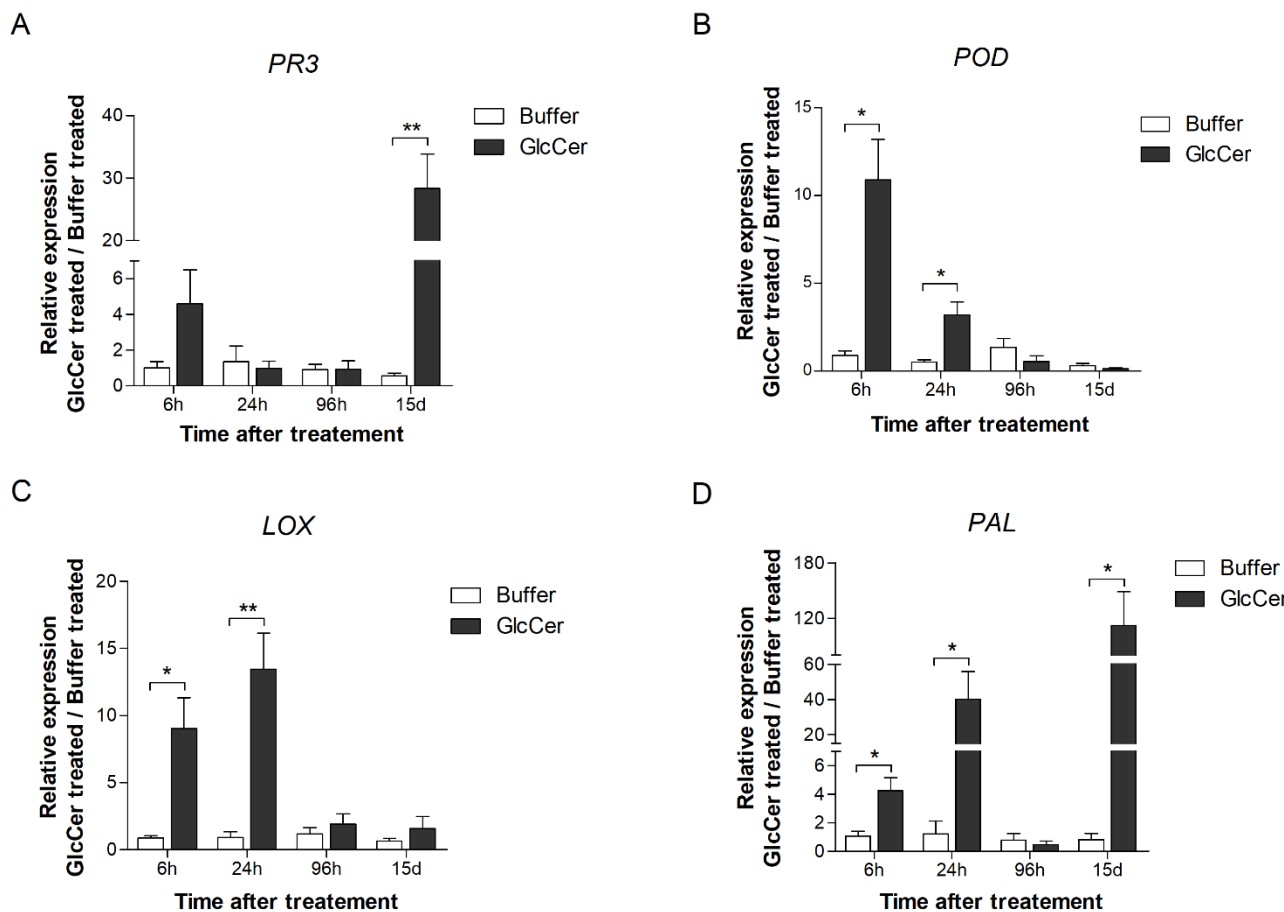

**Figure 7.** Expression of genes related to the plant defense of *P. edulis* treated with *C. herbarum* GlcCer at different time points. (**A**) *PR3* gene analysis (chitinase); (**B**) *POD* gene analysis (peroxidase); (**C**) *LOX* gene analysis (lipoxygenase) and (**D**) *PAL* gene analysis (phenylalanine ammonia lyase). Constitutive genes used for normalization: *NDID* (NADP-dependent isocitrate dehydrogenase) and *ERS* (Ethylene Response Sensor). The analyses were made from the average of two biologically independent experiments and technical triplicate of each biological replicate, and the relative expression was calculated by the $2^{-\Delta\Delta Ct}$ method. Asterisks denote values statistically different from control. * $p < 0.05$, ** $p < 0.01$.

### 3.6. Evaluation of Plant Developmental Parameters

*P. elidus* development was analyzed to investigate whether GlcCer treatment can influence plant growth in late periods after GlcCer stimulation. For this reason, plant height and root weight were measured 40 days after being sprayed. Plants treated with GlcCer showed a height around 80 cm, which is almost 4-fold higher than the untreated or buffer-treated samples that displayed approximately 30 cm of height (Figure 8). Regarding the root weight, GlcCer increased fresh weight from around 1 g in untreated and buffer-treated samples to 4 g in GlcCer-treated plants (Figure 9). Dry weight was also increased from approximately 0.2 g in untreated and buffer-treated samples to 0.6 g in GlcCer-treated plants (Figure 9). These data suggest that GlcCer treatment induced plant growth, as evidenced by the increased plant height and root weight.

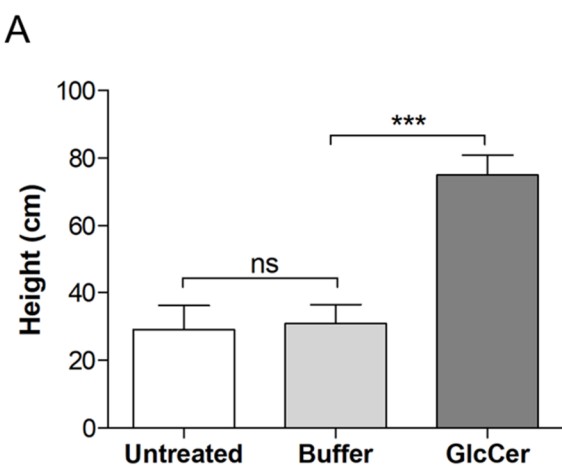
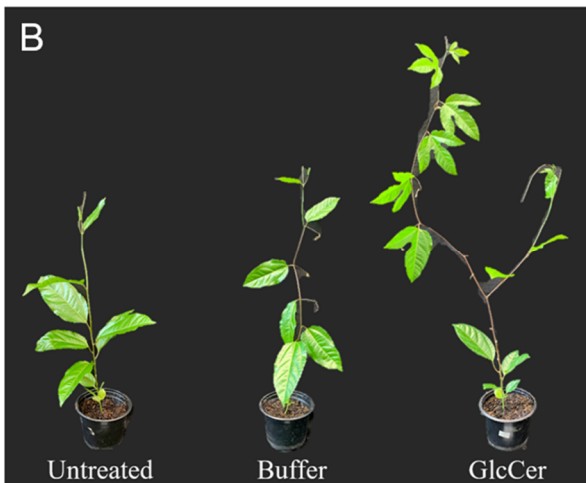

**Figure 8.** Effects of GlcCer species on plant height. (**A**) plant height measured after 40 days after being sprayed. (**B**) Appearance of plants after 40 days after being sprayed. GlcCer (100 µg/mL) was used to spray plants and controls were sprayed with only buffer. Untreated plants were used as negative control. Untreated: plants without treatment; Buffer: plants sprayed with potassium phosphate buffer 0.01 M; and GlcCer: plants sprayed with 100 µg/mL GlcCer. The height of five plants per treatment was determined 40 days after spraying. ns = not significant. *** $p < 0.001$.

## 4. Discussion

Glucosylceramides are conserved molecules that have already been described in a variety of fungi, including not only human pathogens, such as *Candida albicans* and *Cryptococcus neoformans* [6,25], but also phytopathogens, such as *Fusarium oxysporum*, *Fusarium graminearum* and *Penicillium digitatum* [9,26,27]. *Cladosporium* species are known to be plant pathogens and contaminants in the environment, but their GlcCer has only been characterized in *C. resinae* so far [6]. Thus, its characterization in *C. herbarum* remained unclear.

The chemical characterization of *C. herbarum* GlcCer showed two major molecular ions at *m/z* 760 and *m/z* 776. This result demonstrates that *C. herbarum* differs from *C. resinae*, which presents two molecular ions at *m/z* 748 and *m/z* 766, both distinct from those found in *C. herbarum* [6]. On the other hand, *m/z* 760 is shared by *C. herbarum* and *F. oxysporum* [9]. GlcCer has already been described for other fungi, such as *Aspergillus fumigatus*, *Candida albicans* and *Scedosporium* species [5,6]. Comparing *C. herbarum* GlcCer with those found in other fungi, it is possible to identify some conserved characteristics, such as the long chain base of 9-methyl-4,8-sphingadienine as well as the hydroxylated fatty acid. Variations among GlcCer structures are detected in the degree of unsaturation and the length of the fatty acid chain [5].

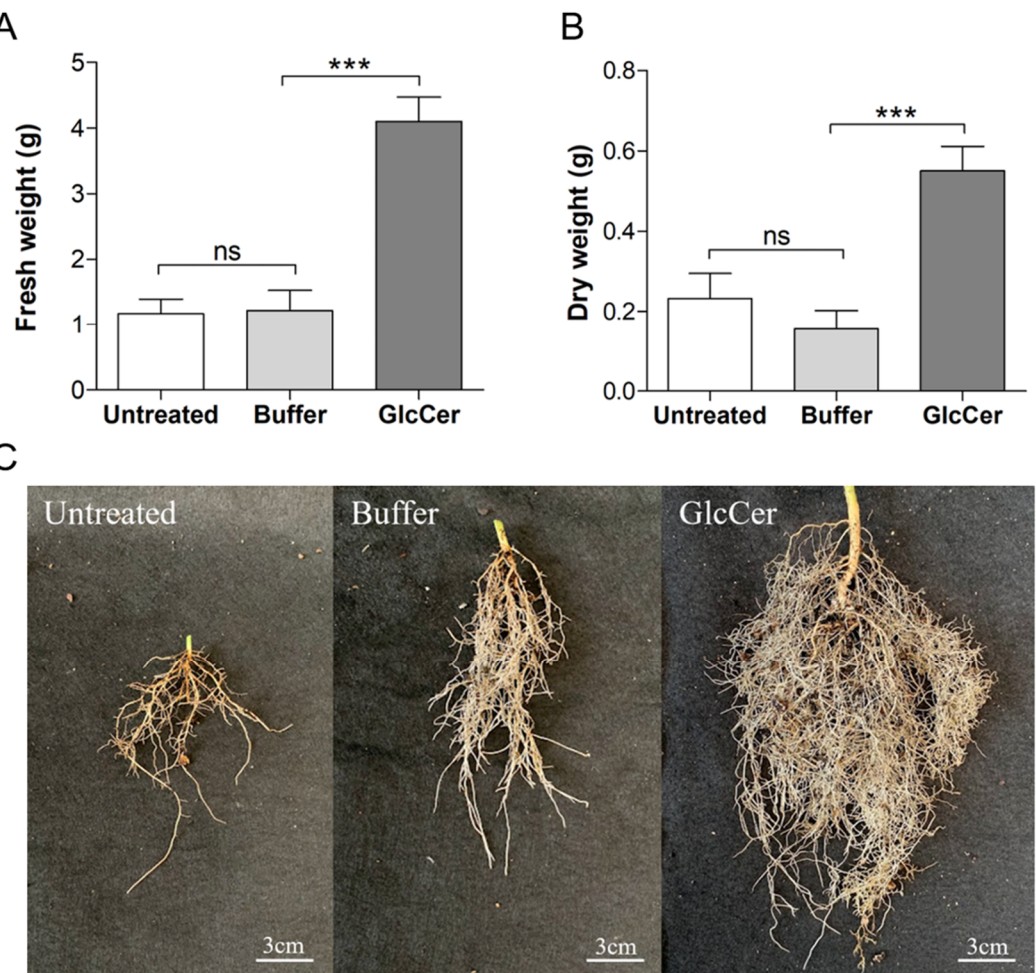

**Figure 9.** Effects of GlcCer on root weight. Untreated plants were used as negative control. (**A**) Root fresh weight. (**B**) Root dry weight. (**C**) Appearance of fresh roots. Untreated: plants with no treatment; Buffer: plants sprayed with potassium phosphate buffer 0.01 M; and GlcCer: plants sprayed with 100 µg/mL GlcCer. The size measurement of five plants for each treatment was made 40 days after spraying. ns = not significant. *** $p < 0.001$.

In *C. herbarum*, GlcCer are exposed on the fungal surface and recognized by monoclonal anti-GlcCer antibodies. These data corroborate those in the literature since GlcCer exposition on the fungal surface has already been observed for *C. neoformans* and *Scedosporium* species, suggesting that it is an important molecule present in the fungal plasma membrane and cell wall [14,16,28–30]. As well as their reactivity to fungal cells, monoclonal antibodies produced with *Aspergillus fumigatus* GlcCer can recognize a variety of purified GlcCer from different fungi [11,16,30,31], including that of *C. herbarum* shown in the present study, suggesting that it contains conserved epitopes among fungal species.

In the present study, we evaluated an alternative for the induction of plant defense mechanisms using purified GlcCer extracted from *C. herbarum* mycelia as an elicitor of the vegetal immune response. The systemic acquired resistance (SAR) has been receiving more attention not only as an alternative for the use of pesticides, but also as a promising prevention treatment against infectious agents. Umemura and colleagues (2004) demonstrated that treatment with a fungal cerebroside induces the production of ROS and the expression of *PR* genes, as well as being able to protect lettuce, tomatoes, melon and sweet potato against infections by *F. oxysporum* [8]. These results indicate that cerebrosides can act as protectors against fungal infections. In addition, Wang and colleagues (2016) showed that the protein PeBA1 extracted from *Bacillus amyloliquefaciens* protects *Nicotiana tabacum*

against Tobacco mosaic virus (TMV) infection, which evidenced the activity of a bacterial molecule against a viral infection [18].

In the present study, we also evaluated whether GlcCer could stimulate plant defense mechanisms. The production of ROS is one of the first lines of plant defense, which plays a crucial role in the recognition of phytopathogens. *C. herbarum* GlcCer treatment led to an increase in *P. edulis* superoxide radicals, suggesting that GlcCer may induce plant defense. This corroborates the work of Umemura and colleagues (2004), which demonstrated that fungal cerebrosides can induce ROS production in sweet potato, melon, lettuce and tomato [8]. In addition, GlcCer from *F. oxysporum* has also been reported to increase ROS production in tobacco, which protects the plant against the infection by TMV [9]. These data indicate that fungal cerebrosides, including GlcCers, play an interesting role in stimulating superoxide release by plants.

Studies showing GlcCers as plant elicitors have been carried out over the last decades [8,9,17,32]. Koga and colleagues (1998) demonstrated that cerebroside degradation products do not have an eliciting activity [32]. Umemura and colleagues (2000) demonstrated that ceramides prepared from cerebrosides, through the removal of glucose, kept on presenting the ability of eliciting defense activity [8]. Furthermore, the methyl group at C-9 and the double bond at the sphingoid base of cerebrosides A and C from *Magnaporthe grisea*, are the key elements that determine the eliciting activity of these compounds in rice [8,32]. In addition, glucocerebrosides from Gaucher's spleen and galactocerebrosides from the bovine brain showed no eliciting activity in rice plants. However, cerebrosides from rice bran showed hardly any activity [8,32].

It has already been observed that the overexpression of certain genes is important for the production of ROS. In 2010, it was reported that *Arabidopsis thaliana* that overexpresses a gene-encoding peroxidase, *RCI3*, presents higher levels of ROS in the roots [33]. Increased ROS production has also been shown in *A. thaliana* overexpressing *CapO2* gene, which encodes an extracellular peroxidase, in response to a bacterium infection [34]. For these reasons, we decided to investigate the expression of some genes involved in plant defense to microbial infections when plants are treated with *C. herbarum* GlcCer.

In the present study, the expressions of *PR3*, *POD*, *LOX* and *PAL* in plants treated with GlcCer were evaluated using RT-qPCR. Gene expression was analyzed after different time points. The *PR3* gene is widely studied due to its direct antifungal activity, which leads to the disruption of components from the inner layer of the fungal cell wall [35]. Our analyses revealed that *PR3* expression was increased at 6 h and 15 days after GlcCer treatment.

The post-formed plant defense is also influenced by molecules from the phenylpropanoid pathway. This metabolic pathway presents the enzyme phenylalanine ammonialyase, which is involved in the production of a variety of phenolic compounds, such as isoflavonoids and some secondary metabolites such as phytoalexins [36]. Our results showed that *PAL* expression was induced by GlcCer at earlier time points, which corroborates some of the data found by Bernardino and colleagues (2020) who observed similar effects using GlcCer from *F. oxysporum* in tobacco plants [9]. In addition, *PAL* expression presented higher levels after 15 days of GlcCer treatment in the present study.

Plant defense is also affected by other genes, such as lipoxygenase gene (*LOX*). The enzyme lipoxygenase (LOX) leads to the formation of high reactivity molecules that induce free radicals and cause damage to the plasma membrane and cell death [37]. On the other hand, the peroxidase gene (*POD*) plays a variety of roles in vegetal defense. POD acts as a mediator of the hypersensitivity response and as a protector against damage caused by ozone and superoxide [38]. In *Arabidopsis*, a mutation in the gene *POD* impairs the development of SAR [39]. The genes associated with plant defense studied in the present work, such as *POD* and *LOX*, showed increased expression at 6 and 24 h after GlcCer treatment, respectively.

The results found in the present study corroborate those described by Naveen and colleagues (2013), who demonstrated that the defense mechanisms of pepper (*Capsicum annum*) are stimulated by cerebrosides extracted from *Colletotrichum capsici*, a fungal pathogen

associated to pepper plants [7]. The molecule was able to induce the production of $H_2O_2$ and the enzymes PAL, POD and LOX, as well as protect the plants against *C. annum* infection. Cerebrosides from *M. grisea* were also demonstrated to be involved in the production of phytoalexins and pathogenesis-related proteins [7,8].

Plant development was evaluated by the determination of plant height and root weight 40 days after GlcCer treatment. Both parameters improved after GlcCer treatment when compared with untreated plants or those treated with buffer. It suggests that GlcCer has a positive effect on plant growth, which is different from some other antiviral compounds that delay plant growth. Similar results were observed when *P. edulis* was treated with a glycoprotein from *C. herbarum* followed by infection with Cowpea aphid-borne mosaic virus (CABMV). The glycoprotein treatment on *P. edulis* led to a decrease in the severity of the disease caused by CABMV when the parameters leaf area, plant height and root weight were evaluated [17]. In addition, tobacco plants treated with GlcCer from *F. oxysporum* also displayed improved growth parameters [9].

GlcCers play many roles in fungal cells, such as fungal growth and differentiation, cellular signaling, extracellular vesicle formation and virulence [5,16,25,40,41]. Therefore, GlcCer from pathogenic fungi were studied during the last decades in order to evaluate its influence in pathogenesis processes. However, little is known about the impacts of GlcCer from phytopathogens on plants and their defense mechanisms. GlcCer from *C. herbarum* led to some alterations in plant development parameters, which resulted in larger plants and larger roots when compared to the control. Similar data have been demonstrated in tobacco treated with GlcCer from *F. oxysporum*, which led to larger and healthier plants compared to the controls [9], suggesting that fungal cerebrosides could help plant growth.

**Supplementary Materials:** The following supporting information can be downloaded at: https://www.mdpi.com/article/10.3390/microbiolres13030028/s1, Table S1: Oligonucleotide primer pairs used for RT-qPCR analysis.

**Author Contributions:** Conceptualization, M.I.D.d.S.X., M.C.B., R.R.-P. and C.B.M.; data curation, M.I.D.d.S.X., M.C.B., R.R.-P., C.B.M. and A.P.d.S.; formal analysis, M.I.D.d.S.X., M.C.B. and R.R.-P.; funding acquisition, M.F.S.V. and E.B.-B.; investigation, M.I.D.d.S.X., M.C.B., R.R.-P., C.B.M., A.P.d.S. and R.O.R.C.; methodology, M.I.D.d.S.X., M.C.B., R.R.-P., C.B.M., A.P.d.S., R.O.R.C. and B.B.M.; project administration, E.B.-B.; resources, M.F.S.V. and E.B.-B.; software, M.I.D.d.S.X., M.C.B., R.R.-P. and B.B.M.; validation, M.I.D.d.S.X., M.C.B., R.R.-P., M.F.S.V. and E.B.-B.; visualization, M.I.D.d.S.X., M.C.B., R.R.-P., R.O.R.C. and B.B.M.; writing—original draft, M.I.D.d.S.X., M.C.B. and R.R.-P.; writing—review and editing, M.F.S.V. and E.B.-B. All authors have read and agreed to the published version of the manuscript.

**Funding:** This research was funded by grants from the Conselho Nacional de desenvolvimento Científico e Tecnológico (CNPq), Fundação de Amparo a Pesquisa do Estado do Rio de Janeiro (FAPERJ) (Xisto, M.I.D.S. was supported by FAPERJ #E-26/202.051/2020; Rollin-Pinheiro, R. was supported by FAPERJ #E-26/200.591/2022), Coordenação de Aperfeiçoamento de Pessoal de Nível Superior (CAPES). The funders had no role in study design, data collection and analysis, decision to publish, or preparation of the manuscript.

**Institutional Review Board Statement:** Not applicable.

**Informed Consent Statement:** Not applicable.

**Data Availability Statement:** Not applicable.

**Acknowledgments:** We thank the Centro de Espectrometria de Massas de Biomoléculas (CEMBIO) from Universidade Federal do Rio de Janeiro for ESI-MS analysis. We thank Jose Leonardo Jiménez Santos for helping with RNA extraction and qPCR procedures. We thank Igor C. Almeida for the critical reading of the manuscript. We also thank Walter Martin Roland Oelemann for the English grammar revision of the manuscript.

**Conflicts of Interest:** The authors declare no conflict of interest.

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
