# Peer review of "Glucosylceramides from Cladosporium and Their Roles in Fungi–Plant Interaction"

_2036-7481, doi:10.3390/microbiolres13030028_

Round 1

Reviewer 1 Report

In this article, Barreto-berger and colleagues describe the isolation and the characterization of glucosylceramides produced by the plant pathogen Cladosporium herbarum. This paper is a continuation of a previous study on GlcCer isolated from C. resinae. Next, the authors investigated the putative role of the GlcCer moiety on the plant-fungal pathogen interaction where they describe an induction of ROS accumulation in leaves and plant defenses by . Surprisingly, GlcCer spray strongly activated plant growth.

The data presented on the chemical characterization of GlcCer species are not correct and are incomplete. In Figure 1, the authors proposed two main structures of GlcCer that are discriminated by a hydroxyl (OH) function. However, the m/z difference is 14, which does not correspond to an OH. Also, the proposed structures contain several double bonds (C=C). However, the analysis of the fragmentation spectra does not allow to determine the positions of hydroxyl group and of C=C double bond. The interpretation appears to be fanciful. Finally, the MS spectrum (Fig. 1A) is very noisy, suggesting that the GlcCer fraction is not pure. In Table 1, same remarks for GlcCer from F. oxysporum. The difference in m/z ion masses between the middle one and the other two does not match the proposed structure. It will be appreciated to follow the fragmentation described by Levery et al. (PMID: 10775088)

The authors need to perform TLC of the GlcCer fraction and other analyses such as NMR and GC-MS to show their purity and to better characterize the structures of GlcCer from C. herbarum.

Effect of GlcCer on plant growth (Figs. 6 and 7). These data are impressive. More details on the protocol will be appreciated. How many sprays/day? Total volume used per spray? .... Does spraying glucose at 100 µg/ml give the same effect?   From my perspective, these data suggest more of an artifact due to sample impurity rather than specific GlcCer activity from C. herbarum. Plants also produce GlcCERs, which are present in several parts (leaves, pollen, roots...). It will be interesting to compare the activity of the two GlcCer species from fungi and plants.

Author Response

Dear Reviewer,

We appreciated all your comments and to attend your interesting suggestions we made several modifications in the manuscript. Please find below a point-by-point answer to all your comments.

  1. The data presented on the chemical characterization of GlcCer species are not correct and are incomplete. In Figure 1, the authors proposed two main structures of GlcCer that are discriminated by a hydroxyl (OH) function. However, the m/z difference is 14, which does not correspond to an OH.

Answer: We are very sorry for the misinterpretation of the GlcCer structures. We repeated the MS analysis and confirm that m/z 760 represents a GlcCer presenting a hydroxylated and unsaturated C18 fatty acid, as already described in other fungal GlcCers. However, the second major molecular ion is in fact m/z 776 (and not m/z 774 as described in the manuscript), which displays an extra hydroxyl group in sphingoid base, indicating the difference of 16 between the two main structures found. New figures (Figures 2 and 3) were elaborate to show these results and the text concerning the characterization of these GlcCer structures was modify in the revised version to explain these differences (Pages 4 and 5).

  1. Also, the proposed structures contain several double bonds (C=C). However, the analysis of the fragmentation spectra does not allow to determine the positions of hydroxyl group and of C=C double bond. The interpretation appears to be fanciful.

Answer: We do apologize for not explaining it properly. According to previous work on the structural characterization of neutral glycosphingolipids from Fusarium species (Biochimica et Biophysica Acta 1390: 186–196, 1998), the m/z increment of 26 between m/z 562 and 536 is consistent with the double bond of 2-hydroxyoctadenenoic acid being in position 3. Similar structure was also found in Aspergillus fumigatus and A.niger.

  1. Finally, the MS spectrum (Fig. 1A) is very noisy, suggesting that the GlcCer fraction is not pure. In Table 1, same remarks for GlcCer from F. oxysporum. The difference in m/z ion masses between the middle one and the other two does not match the proposed structure. It will be appreciated to follow the fragmentation described by Levery et al. (PMID: 10775088)

Answer: We apologize for the MS spectrum. And a new figure (Figure 1) was included in the revised manuscript in order to show the purity of the glucosylceramide fraction used in the biological studies. Also, a new figure (Figure 3) showing the fragmentation of the major ions at m/z 760 and m/z 746 following the fragmentation described by Levery et al (Rapid Commun. Mass Spectrom. 14, 551–563, 2000) and based in the nomenclature of Costello et al, as modified by Adams and Ann (Adams J, Ann Q. Mass Spectrom. Rev. 1993; 12: 51) was used in the revised version of the manuscript (Figure 3).

In addition, we agree with the referee comment concerning the GlcCer structures from F. oxysporum showed in Table 1.  The proposed structure for the molecular ion specie at m/z 776 is wrong. An Δ3 unsaturation in the fatty acid chain is missing. For this reason, we decided to remove the Table 1 from the revised manuscript to avoid misunderstandings.

  1. The authors need to perform TLC of the GlcCer fraction and other analyses such as NMR and GC-MS to show their purity and to better characterize the structures of GlcCer from C. herbarum.

Answer: Thank you for your suggestion. A new figure (Figure 1) showing the strategy used for purification of GlcCer from Cladosporium herbarum as well as high-performance thin-layer chromatography of neutral glycosphingolipids isolated by silica gel column chromatography, which indicates that the GlcCer used in the study is a pure fraction obtained from C. herbarum.

  1. Effect of GlcCer on plant growth (Figs. 6 and 7). These data are impressive. More details on the protocol will be appreciated. How many sprays/day? Total volume used per spray? .... Does spraying glucose at 100 µg/ml give the same effect?   From my perspective, these data suggest more of an artifact due to sample impurity rather than specific GlcCer activity from C. herbarum.

Answer: More details of the protocol description were inserted in the Materials and Methods section: Page 3, title “2.8. GlcCer pulverization on P. edulis”: “The passion fruit plants were sprayed only once and the leaves were collected after 6 h, 24 h, 96 h and 15 days for gene expression evaluation or for histochemical detection of reactive species oxygen (ROS). The volume per spray for each young plant was about 5 ml. The plant developmental evaluation was performed as described for the gene expression evaluation and ROS detection.”

            We did not spray glucose at 100 µg/ml to know if it would give the same effect, since it has been demonstrated that ceramides prepared from cerebrosides, through the removal of glucose, kept on presenting the ability of eliciting defense activity (Umemura et al. 2000). For better explanation, we inserted a paragraph on Discussion Section, Page 15: “Koga and colleagues (1998) have demonstrated that cerebroside degradation products do not have an eliciting activity (Koga et al. 1998). Umemura and colleagues (2000) demonstrated that ceramides prepared from cerebrosides, through the removal of glucose, kept on presenting the ability of eliciting defense activity (Umemura et al. 2000).” 

  1. Plants also produce GlcCers, which are present in several parts (leaves, pollen, roots...). It will be interesting to compare the activity of the two GlcCer species from fungi and plants.

Answer: It has been demonstrated that mammalian and plant cerebrosides do not show elicitor activity in plants, since the methyl group at C-9 and the double bond at the sphingoid base of fungal cerebrosides are the key elements that determine the eliciting activity in plant, and these cerebroside structure are absent in plants.

For better explanation, we inserted a paragraph on Discussion Section, Page 15: “Studies showing GlcCers as plant elicitors have been carried out over the last decades (Koga et al. 1998; Umemura et al. 2000, Bernardino et al. 2020, Santos-Jiménez et al. 2022). Koga and colleagues (1998) have demonstrated that cerebroside degradation products do not have an eliciting activity (Koga et al. 1998). Umemura and colleagues (2000) demonstrated that ceramides prepared from cerebrosides, through the removal of glucose, kept on presenting the ability of eliciting defense activity (Umemura et al. 2000).  Furthermore, the methyl group at C-9 and the double bond at the sphingoid base of cerebrosides A and C from Magnaporthe grisea, are the key elements that determine the eliciting activity of these compounds in rice (Koga et al. 1998; Umemura et al. 2000). In addition, glucocerebrosides from Gaucher’s spleen and galactocerebrosides from the bovine brain showed no eliciting activity in rice plants. However, cerebrosides from rice bran showed hardly any activity (Koga et al. 1998; Umemura et al. 2000).”

Reviewer 2 Report

This paper describes a study on the chemical structure of glucosylceramides (GlcCer) from the fungus Cladosporium herbarum. Using monoclonal antibodies the authors  show that these molecules are exposed on the fungal surface. In addition they found that  treatment of Passiflora edulis with GlcCer induced increased levels of superoxide as well as the expression of several   genes related to plant defense. Finally they show that  GlcCer treatment  also enhanced several growth parameters. All experiments are well described, with the appropriate controls and consequently the results are significant. This paper adds to the increasing knowledge on fungal cerebrosides, their structure and possible role in the relation between fungi, plants and plant pathogens. The paper is well written.

Author Response

Dear Reviewer,

Thank you, we appreciate your revision and we are happy that you like the manuscript.

Round 2

Reviewer 1 Report

The revised manuscript of Barreto-berger et al. partially addresses my comments. There are still some problems with the MS analysis. In Figure 3, the daughter ion at m/z = 578 in the scheme B is missing in the MS spectra of panel A. In panel D, the daughter ion at m/z = 524 is in the wrong place in the scheme D (it should be between C=O and CH-OH). The position of the fatty acid double bond is not in the same location between schemes B and D. There is no data from the mass spectra to discriminate their position. Finally, there are no MS daughter ion to indicate the position of the additional hydroxyl group on the sphingadienine base in schemel D. As previously requested, GC-MS analysis of the hydrolyzed sample is required to establish the structure of the two major GlcCer structures.

The authors did not address the effect of the GlcCer fraction on plant growth and a putative artifact. Does spraying glucose at 100 µg/ml give the same effect?

Author Response

Dear Reviewer,

Thank you for your comments. Please see below a point-by-point answer for your suggestions.

1) The revised manuscript of Barreto-berger et al. partially addresses my comments. There are still some problems with the MS analysis. In Figure 3, the daughter ion at m/z = 578 in the scheme B is missing in the MS spectra of panel A. In panel D, the daughter ion at m/z = 524 is in the wrong place in the scheme D (it should be between C=O and CH-OH). The position of the fatty acid double bond is not in the same location between schemes B and D. There is no data from the mass spectra to discriminate their position. Finally, there are no MS daughter ion to indicate the position of the additional hydroxyl group on the sphingadienine base in schemel D. As previously requested, GC-MS analysis of the hydrolyzed sample is required to establish the structure of the two major GlcCer structures.

Answer:

Figure 3 – daughter ion at m/z 578: As the daughter ion at m/z is barely detected in the MS spectrum of panel A, we removed it from scheme B (Page 7).

Panel D: The daughter ion at m/z 524 is now in the right place in the scheme D (Page 7).

Position of the fatty acid double bond:  The position of the fatty acid double bond in both GlcCer structures is in the same location. And is consistent with the double bond of 2-hydroxyoctadecenoyl acid being in position 3 (BBA, 1390:186-196,1998). It was corrected in Scheme D (Page 7).

Position of the additional hydroxyl group on LCB: Based on ESI-MS (Collision activation of protonated and lithium cationized method) and sugar analysis, we proposed that the structure of the major glycosphingolipid from C. herbarum (m/z 760) is N-2`-hydroxyoctadecenoyl-1-O-β-Dglucopyranosyl-9-methyl-4,8-sphingadienine. However, the glycosphingolipid of m/z 776   contains a 9-methyl-4,8-sphingadienine carrying an extra hydroxyl group. The difference of 16 units observed among the GlcCer parental ions (Figure 2 A) remained detectable after loss of either monosaccharide or fatty acid, suggesting that the long chain base could possibly present an extra hydroxyl group. In a previous paper (Infect. Immun. 2005, 73(12):7860), this hypothesis was supported after analysis of the permethylated GlcCer derivative. Fragmentation analysis suggests that the additional hydroxyl group is linked to C-4 or C-5 of its long chain base. Based on these results we proposed the structure for the molecular species with m/z 776 (Panel D). However, the proposed structure was based on the similarity of the fragments as explained in the text. So, for a better understanding we modified the structure and in the new figure the hydroxyl group in the 9-methyl- 4,8-sphingadienine is shown in an unspecified location in parenthesis (new scheme D) (Pages 5 and 7).

GC-MS analysis of the hydrolysed sample: Unfortunately, we have no conditions at the moment to perform a GC-MS analysis of the major GlcCer from C. herbarum. The proposed structures were based on ESI-MS analysis of the lithiated molecules, HPTLC for the sugar identification and the literature data on the structures of similar fungal glycosphingolipids.

2) The authors did not address the effect of the GlcCer fraction on plant growth and a putative artifact. Does spraying glucose at 100 µg/ml give the same effect?

Answer:

We did not use glucose alone as a control. Based on the literature, other studies that evaluate the influence of lipids in plant parameters also did not use glucose as a control. According to the purification analysis (Figure 1), the sample is pure and the effect found on plants is specific to the GlcCer. In addition, data in the literature indicate that lipid molecules, such as sphingolipids, can influence plant growth and mechanisms of defense, as we discussed in the following paragraph: “Studies showing GlcCers as plant elicitors have been carried out over the last decades [8, 9, 17, 32]. Koga and colleagues (1998) have demonstrated that cerebroside degradation products do not have an eliciting activity [32]. Umemura and colleagues (2000) demonstrated that ceramides prepared from cerebrosides, through the removal of glucose, kept on presenting the ability of eliciting defense activity [8].  Furthermore, the methyl group at C-9 and the double bond at the sphingoid base of cerebrosides A and C from Magnaporthe grisea, are the key elements that determine the eliciting activity of these compounds in rice [8, 32]. In addition, glucocerebrosides from Gaucher’s spleen and galactocerebrosides from the bovine brain showed no eliciting activity in rice plants. However, cerebrosides from rice bran showed hardly any activity [8, 32].”

These mentioned data indicate that the effect on plants depends on the lipid structure, which suggests that GlcCer did not act as an artifact in the experiments.
